# DUAL-TARGET POINT CLOUD REGISTRATION USING REPRESENTATIVE OVERLAPPING POINTS

## ABSTRACT

Point cloud registration is a challenging task when only partially visible data is available. Recently, many learning-based methods have been proposed for this problem and have achieved satisfactory performance. However, they rarely combine multiple features and fail to pay attention to the key factor of registration: alignment of attitude. Based on this phenomenon, we propose a dual-target point cloud registration model, which combines multiple features learned from Point-Net, DGCNN, and attention module. First, an initialization module is introduced for coarse registration, in which a new target point cloud is added compared to the original model. Second, we design a two-step attention-based representative overlapping-point selection module, which uses both global and local features of input point clouds. In the first step, overlapping scores are predicted using global features. In the second step, a feature-matching matrix is obtained based on local features and a self-attention module. Representative overlapping points are selected based on the overlapping scores in the first step and the feature-matching matrix in the second step. Finally, a weighted SVD algorithm is used to estimate the transformation from the point cloud after initialization to the target point cloud. Extensive experiments on ModelNet40 show our method achieves state-of-the-art performance compared to other learning-based methods. The code is available at https://github.com/Dual-target.

## 1 INTRODUCTION

Point cloud registration is a fundamental task in robot vision applications, such as automatic driving(Qin et al., 2020; Wang et al., 2023), 3D reconstruction(Wu et al., 2023a; Zhou & Tulsiani, 2023), and robot localization (Yin et al., 2021; Feng et al., 2023). The problem becomes even harder when there is no exact correspondence between input point clouds, owing to the non-overlapping points and noise. To solve this problem, many algorithms have been proposed in recent years.

Iterative Closest Point (ICP)(Besl & McKay, 1992) is the most classical method for point cloud registration, which iteratively alternates between two sub-problems: finding the closest points and computing optimal rigid transformation. Although ICP is straightforward to implement and can achieve adequate results in some scenes, it can only converge to a local optimum near the initial position. To solve the non-convexity of ICP, Go-ICP(Yang et al., 2016) was proposed, which bypasses the local optimum using the branch-and-bound algorithm. However, its time cost is prohibitive. Worse still, ICP and its derivative algorithms(Yang et al., 2016; Rusinkiewicz, 2019) fail in partial-to-partial point cloud registration problem, because they obtain correspondences based on the closest points between input point clouds and fail to notice the existence of noise and non-overlapping points. This will lead to erroneous correspondences without doubt.

Recently, learning-based methods(Wang & Solomon, 2019b; Yuan et al., 2020; Qin et al., 2022; Wu et al., 2023b) have shown promising results for solving this problem. PointNetLK(Aoki et al., 2019) and DCP(Wang & Solomon, 2019a) use the PointNet, dynamic graph convolutional neural network(DGCNN) or transformer to extract the features of input point clouds, showing good robustness against noise. However, these algorithms still cannot perform well in partial-to-partial point cloud registration problem, because a simple PointNet(Charles et al., 2017) or cross-attention mechanism network can hardly learn the features of non-overlapping points. Therefore, RPMNet(Yew & Lee, 2020) and IDAM(Li et al., 2020) were proposed successively. However, they extract reliable

features only when there are distinctive local geometric structures. Recently, ROPNet(Zhu et al., 2021) selects representative overlapping points from the source point cloud, converting this problem into a partial-to-complete point cloud registration problem. However, ROPNet ignores local features and the combination of multi-level features. FINet(Xu et al., 2022) solves this problem from a new perspective, by introducing multi-level feature interactions between the input point clouds. Besides, DIFTChen et al. (2023) introduces point cloud structure extractor and point feature transformer module, measuring spatial consistency and estimating correspondence confidence based on geometric matching, which improves the accuracy of correspondences. However, all of them fail to pay attention to the key factor determining the accuracy of registration: alignment of attitude.

The difficulty in point cloud registration lies in aligning the point cloud attitude. So in this paper, we deal with the problem from a new perspective, by introducing a dual-target point cloud registration model. We have shown adding a new target point cloud obtained by rotation from the source point cloud, can improve the accuracy of registration. The newly added target point cloud has the same attitude as the target point cloud. The spatial locations of source, target, and newly added point clouds are shown in the figure of Point Cloud Pair in Fig. 1. Besides, global features(Charles et al., 2017) capture the overall shape and distribution information of the point cloud, while local features(Wang et al., 2019) include the specific structure and local details of the point cloud. Therefore, both global features and local features are needed for point cloud registration.

Based on the above discussion, we design a dual-target point cloud registration model which integrates multiple features to solve the partial-to-partial point cloud registration problem. First, we introduce a Res-PointNet module for initial transformation. Compared to PCRNet(Sarode et al., 2019), our Res-PointNet can aggregate global features from multi-level. Second, we design an Attention-based Representative Overlapping-Point Selection(AROPS) module which uses the transformed point cloud completely. The AROPS module is divided into two parts, one uses the global features based on PointNet and the other uses the local features based on DGCNN(Wang et al., 2019). In the first part, a multiple information fusion module is used to predict overlapping scores and select preliminary overlapping points, where the information includes the high dimensional point features, the multi-level global features and corresponding structure information encoded by a transformer module. In the second part, an Attention-based Mismatched-Point Removal(AMPR) module is used to get the final representative overlapping points and the similarity matrix. Finally, a weighted SVD algorithm is used to estimate the transformation from the point cloud after initialization to the target point cloud. Combining the transformation in the initialization module and the Attention-based Representative Overlapping-Point Selection module, the finally transformation can be calculated. The pipeline of our model is shown in Fig. 1.

## 2  RELATED WORKS

**Global Feature-based Methods.**   Since PointNet(Charles et al., 2017) was proposed, many registration models based on global features have emerged, and PointNetLK(Aoki et al., 2019) is the pioneer among them. PointNetLK utilizes the Lucas & Kanade algorithm(Lucas & Kanade, 1981) to handle the registration problem. PCRNet(Sarode et al., 2019) replaces the LK algorithm with a MLP network and transforms the rigid transformation problem into a fitting problem. FINet(Xu et al., 2022) introduces multi-level feature interactions between the input point clouds. However, it is difficult for these methods that rely solely on global features to learn the features of non-overlapping points. Besides, because they are based on PointNet(Charles et al., 2017), local features will inevitably be ignored.

**Correspondence Matching-based Methods.**   ICP(Besl & McKay, 1992) is the most classical correspondence matching-based method, which calculates correspondence based on the closest points. However, due to the non-convexity of ICP, the algorithm can only converge to a local minimum near the initial position. Although ICP's variants including GoICP(Yang et al., 2016) and Symmetric ICP(Rusinkiewicz, 2019) alleviate this shortcoming to a certain extent, none of them notice the partially visible data. Recently learning-based methods replace handcrafted descriptors with MLP, GNN or attention mechanisms, showing promising results for solving partial-to-partial point cloud registration problem. PRNet(Wang & Solomon, 2019b) and RPMNet(Yew & Lee, 2020) are early works that pay attention to the partially visible data. They use Gumble-Softmax(Maddison et al., 2016) and Sinkhorn normalization(Sinkhorn & Knopp, 1967) to improve the accuracy of

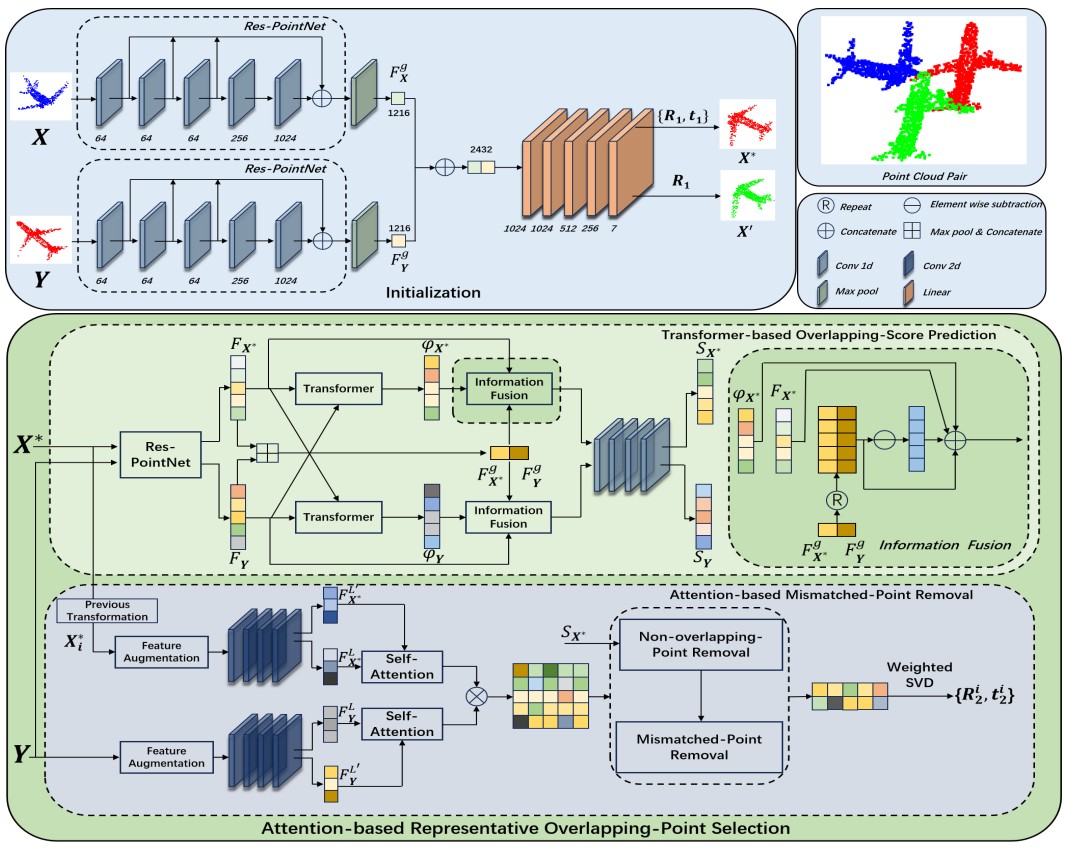

Figure 1: Network architecture of our model, including the Initialization module(top) and the Attention-based Representative Overlapping-Point Selection module(bottom). The blue, red, and green point clouds in the figure of Point Cloud Pair are the source, target and newly added target point clouds respectively.

feature-matching. OMNet(Xu et al., 2021) estimates overlapping masks for non-overlapping points. Similarly, ROPNet(Zhu et al., 2021) predicts overlapping scores to select representative points. OGMM(Mei et al., 2023) proposes a novel overlap-guided probabilistic registration approach, reformulating the problem as aligning two Gaussian mixtures. Nevertheless, all of them use a single feature and ignore the key factor of registration.

**Loss function.** For learning-based methods, the choice of loss function will affect the quality of their models. DCP(Wang & Solomon, 2019a) measures the deviation of $(R, t)$ from the ground truth. In OMNet(Xu et al., 2021) and FINet(Xu et al., 2022), $R$ is replaced by the quaternion. PCRNet(Sarode et al., 2019) uses a loss function called Earth Mover Distance(Rubner et al., 2000), which minimizes the distance between the corresponding points of source and target point clouds and is robust to outliers. ROPNet(Zhu et al., 2021) replaces Earth Mover Distance with a simple function to calculate the absolute error between source and target point clouds. However, none of them pay attention to the key of registration: alignment of attitude. We add a new target point cloud obtained by rotation from the source point cloud to increase the weight of the rotation in the loss function. It is proven in our paper that rotation-preference loss function is better than the original.

## 3 METHOD

In our model, the transformation from the source point cloud to the target point cloud is divided into two steps. The first step is an initialization model for coarse registration. The second step is an accurate transformation from the point cloud after initialization to the target point cloud.

## 3.1 Initialization

As shown in ROPNet(Zhu et al., 2021), an initial transformation is favored by predicting overlapping points. Considering the model complexity, the initialization module should be as simple as possible, but still be effective. PCRNet(Sarode et al., 2019) uses a PointNet style network to extract global features of the input point clouds and regresses a quaternion and a translation vector. However, PCRNet regresses the transformation using the high-level global features only and loses sight of the low-level features, which are also exactly what point cloud registration needs. To combine high-level and low-level global features, we introduce a module called Res-PointNet which concatenates the output of each convolution block in the MLPs, except for the medium-level convolution block. The combination of high-level and low-level features is formulated as:

$$
\begin{aligned}
F_X &= \text{cat}\left(h_1(X_0), h_2\left(X_1\right), h_3\left(X_2\right), h_5\left(X_4\right)\right), \\
F_Y &= \text{cat}\left(h_1(Y_0), h_2\left(Y_1\right), h_3\left(Y_2\right), h_5\left(Y_4\right)\right),
\end{aligned}
\tag{1}
$$

where $F_X$ and $F_Y$ respectively denote the combination of high-level and low-level features of the source point cloud $X$ and target point cloud $Y$. The symbol $cat(\cdot)$ represents concatenation. Operators $h_i$ is the $i$th convolution block. The symbols $X_i$ and $Y_i$ are the outputs of the $i$th convolution block. Here, the outputs of fourth convolution block are not used. Follow PointNet, a symmetric function max-pooling $max(\cdot)$ is used to get global features $F_X^g$ and $F_Y^g$. The initial transformation is formulated as:

$$
\text{v} = \text{h}_\theta\left(\text{cat}\left(F_X^g, F_Y^g\right)\right),
\tag{2}
$$

where $\text{h}_\theta$ denotes the transformation decoder, which is a simple fully connected layer. The output v is a 7-dimensional vector whose first four values denote the quaternion and the last three values represent a translation vector. Therefore, the transformed source point cloud $X^*$ after initial transformation can be formulated as $X^* = X \cdot \text{R}_1 + \text{t}_1$, and the transformation to the newly added target point cloud is formulated as $X' = X \cdot \text{R}_1$, where $\text{R}_1$ is obtained in terms of the quaternion.

## 3.2 Attention-based Representative Overlapping-Point Selection

The Attention-based Representative Overlapping-Point Selection(AROPS) module is established on the point cloud after the initial transformation. To achieve this method, we first design a Transformer-based Overlapping-Score Prediction(TOSP) module, which combines multiple features to calculate overlapping scores and selects overlapping points preliminarily. Besides, an Attention-based Mismatched-Point Removal(AMPR) module is used to remove mismatched points with wrong correspondences from these overlapping points we have selected based on overlapping scores. Finally, the transformation from the point cloud after initialization to the target point cloud can be estimated by a weighted SVD algorithm(Papadopoulo & Lourakis, 2000).

### 3.2.1 Transformer-based Overlapping-Score Prediction

There are some non-overlapping points between the input point clouds in the task of partial-to-partial point cloud registration. In our model, a module called Transformer-based Overlapping-Score Prediction(TOSP) is designed to predict overlapping scores and remove these non-overlapping points from the transformed source point cloud $X^*$.

**Transformer.** Since transformer(Vaswani et al., 2017; Han et al., 2023) was proposed, it has been widely used in various deep learning tasks. The main feature of transformer is that it uses the multi-head self-attention mechanism to capture long-distance dependencies in sequences, which convolution block cannot do. Besides, in the field of 3D point cloud processing, transformer can capture the internal geometry and correlation information of the input point cloudsGuo et al. (2021); Engel et al. (2021); Qin et al. (2023). Inspired by the success of BERT(Devlin et al., 2019) and DCP(Wang & Solomon, 2019a), a module called inter-transformer is used to learn the correlation between source and target point clouds by self-attention and conditional attention mechanism.

To reduce network parameters, we share convolution kernel parameters with the Res-PointNet used in the initialization module. Take $F_{X^*}$ and $F_Y$ to be the embedding generated by the Res-PointNet. The inter-transformer module is dedicated to learning a mapping : $\Phi : \mathbb{R}^{N \times C} \times \mathbb{R}^{N \times C} \to \mathbb{R}^{N \times C}$,

where $C$ is the output dimension of the Res-PointNet module. Then the new embedding of the transformed source and target point clouds is formulated as:

$$
\begin{aligned}
\varphi_{X^*} &= \Phi\left(F_{X^*}, F_Y\right), \\
\varphi_Y &= \Phi\left(F_Y, F_{X^*}\right),
\end{aligned}
\tag{3}
$$

where $\varphi_{X^*}$ and $\varphi_Y$ are new embedding of the transformed source and target point clouds. Considering the embedding $F_{X^*}$ and $F_Y$ will be used later, so there is no a residual network as DCP(Wang & Solomon, 2019a). The inter-transformer module is the same as the attention module in DCP, which is composed by several stacked encoder-decoder layers. The aim of these encoder layers is to encode $F_{X^*}$ or $F_Y$ to an embedding space. The aim of these decoder layers is to encode $F_Y$ or $F_{X^*}$ and relate the embedding of $F_Y$ or $F_{X^*}$ to the other embedding encoded in the encoder layers.

**Information Fusion.** We introduce an information fusion module to predict overlapping scores based on global features and corresponding structure information obtained by the transformer module. For a more accurate prediction for overlapping scores, this module is used after initialization. In our model, the task of predicting overlapping scores is a binary classification problem. The overlapping scores can be calculated by :

$$
\begin{aligned}
S_{X^*} &= h_\psi\left(\mathrm{cat}\left(F_{X^*}, r\left(F_{X^*}^g\right), r\left(F_Y^g\right), r\left(F_{X^*}^g - F_Y^g\right), \varphi_{X^*}\right)\right), \\
S_Y &= h_\psi\left(\mathrm{cat}\left(F_Y, r\left(F_Y^g\right), r\left(F_{X^*}^g\right), r\left(F_Y^g - F_{X^*}^g\right), \varphi_Y\right)\right),
\end{aligned}
\tag{4}
$$

where $h_\psi$ denotes the overlapping decoder that is a PointNet style network with a soft-max function. And $r(\cdot)$ is a function of expanding dimensions by repeating features. The $S_{X^*}$, $S_Y$ are the overlapping scores of $X^*$ and $Y$. Through this module, non-overlapping points will be removed and a new transformed source point cloud $X_{ro1}^*$ is obtained. The overlapping decoder $h_\psi$ contains 5 Conv1d(2048,2048,512,256,2) with $Relu$ and group normalization except the last block.

### 3.2.2 Attention-based Mismatched-Point Removal

After removing these non-overlapping points, there are still some points in the point cloud $X_{ro1}^*$ with a wrong correspondence to the points in the point cloud $Y$. Therefore, a mismatched-point removal module is used to remove these points in $X_{ro1}^*$ with wrong correspondences to points in $Y$. Mismatched points are removed based on the point cloud $X_{ro1}^*$, however, the features are extracted based on the transformed point cloud $X^*$ with non-overlapping points.

**Point-Feature Augmentation.** To enrich point features, a Point-Feature Augmentation module is used following ROPNet(Zhu et al., 2021). For each point $x^* \in \mathbb{R}^3$ in the transformed point cloud $X^*$, its k-nearest neighbor points $M_k \in \mathbb{R}^{k \times 3}$ are selected. Besides, the spatial coordinate of each point and the Point-Pair-Feature(PPF)(Drost et al., 2010) are concatenated to it together. The Point-Pair-Feature(PPF) contains the normal information of the point cloud and is represented by symbol $F^P$ in our paper. Therefore, the augmented feature $F_{m_i}^L \in \mathbb{R}^{10}$ of the point $x_i^*$ is formulated as :

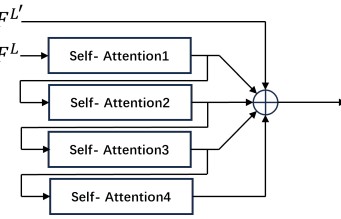

Figure 2: Architecture of the Attention layer.

$$
F_{m_i}^L = cat(m_i, m_i - x, F_i^P).
\tag{5}
$$

Local features can capture the geometric structure and local details of the point cloud, which is beneficial for improving the robustness in point cloud processing(Qi et al., 2017; Liu et al., 2019; Lin et al., 2023). Therefore, before self-attention, a Dynamic Graph Convolutional Neural Network(DGCNN)(Wang et al., 2019) is used with group normalization(Wu & He, 2018) instead of batch normalization(Ioffe & Szegedy, 2015) to encode the embedding $F_{m_i}^L$ to a high-dimensional space. The output $F_{X^*}^L$ is formulated as :

$$
F_{X^*}^L = \max_{1 \le i \le k}\left(\mu_\theta\left(F_{m_i}^L\right)\right),
\tag{6}
$$

where $\max_{1 \le i \le k}(\cdot)$ denotes a max-pooling function, and $\mu_\theta(\cdot)$ is a convolution layer which is made up of several 2d-convolution blocks. The same operation applies to the target point cloud $Y$.

**Self-Attention.** After Point-Feature Augmentation with DGCNN, the embedding has learned the local-feature representation for the point clouds. However, the global dependencies of the point clouds are also needed. Therefore, several self-attention blocks are used, which can learn the distance, relative position, and correlation degree between points, and capture the long-distance correlation among points in the point cloud. The network architecture of our self-attention layer is shown in Fig. 2. In contrast to $F_{X^*}^L$ and $F_Y^L$, $F_{X^*}^{L'}$ and $F_Y^{L'}$ are the outputs of DGCNN without $ReLU$ in the last convolution block. Given the augmented feature $F_{X^*}^L$ for the transformed point cloud $X^*$, the self-attention layers can be formulated as:

$$Q_i = F_{X^*}^{L,i} \cdot W_i^Q, \quad K_i = F_{X^*}^{L,i} \cdot W_i^K, \quad V_i = F_{X^*}^{L,i} \cdot W_i^V,$$
$$F_{X^*}^{L,i+1} = \text{norm}\left(\text{softmax}\left(Q_i \cdot K_i^T\right)\right) \cdot V_i. \tag{7}$$

There are several self-attention blocks, where $i$ is the index of the self-attention block. The input $F_{X^*}^{L,i}$ of $i$th self-attention block is the output of the previous one. Up to here, the embedding has learned the relative position and correlation degree between points. However, local features may be forgotten after these self-attention blocks. Therefore, the augmented features $F_{X^*}^{L'}$ and $F_Y^{L'}$ are concatenated to the output of each self-attention block, where the augmented features $F_{X^*}^{L'}$ and $F_Y^{L'}$ are the embedding of DGCNN without $ReLU$ in the last conv-2d block. The final embedding of the transformed source and target point clouds can be represented as $F_{X^*}^t \in \mathbb{R}^{N \times 5C_P}$ and $F_Y^t \in \mathbb{R}^{N \times 5C_P}$, $C_P$ is the dimension of the output of each attention block, which is 256 in our paper.

**Representative Overlapping-Point Selection.** These representative overlapping points are selected through a two-step process followed ROPNet(Zhu et al., 2021). In the first step, non-overlapping points are removed and then $N_1$ overlapping points are left as the first batch overlapping point cloud $X_{ro1}^* \in \mathbb{R}^{N_1 \times 3}$ on the basis of overlapping scores $S_{X^*}$. In the second step, we further remove mismatched points to obtain the final $N_2$ points as the representative overlapping point cloud $X_{ro2}^* \in \mathbb{R}^{N_2 \times 3}$. The second step can be formulated as:

$$X_{ro2}^* = X_{ro1}^* \left[ \underset{\text{top-prob}}{\arg\max} \left( \max_j \left( F_{X_{ro1}^*}^t \cdot F_Y^{t\,T} \right)_{ij} \right) \right], \tag{8}$$

where top-prob means selecting a certain number of indexes based on the probability distribution of similarity scores calculated by the similarity matrix $H_{ro1} = F_{X_{ro1}^*}^t \cdot F_Y^{t\,T}$. So far the finally representative overlapping points are obtained. Considering the final similarity matrix $H_{ro2} = F_{X_{ro2}^*}^t \cdot F_Y^{t\,T}$, we can match the corresponding points in $Y$ for each point $x_{ro2}^i$ in the representative source point cloud $X_{ro2}^*$ as follows:

$$\mathbb{J} = \underset{\text{top}-k}{\arg\max} H_{ro2,i}^*, \quad w_{ij} = \begin{cases} H_{ro2,ij}^* & j \in \mathbb{J} \\ 0 & j \notin \mathbb{J} \end{cases}, \quad y_i = \frac{w_{i,:}}{\sum_j w_{i,j}} \cdot Y. \tag{9}$$

Where $J$ denotes the indices of $Y$ which have $top - k$ maximum similarity scores for $x_{ro2}^*$. The corresponding point pairs $(x_{ro2,i}, y_j)$ can be obtained based on the formula above. Based on the weight $S_{X^*,i}$, a weighted SVD algorithm will be used to estimate the transformation $R_2$ and $t_2$ from $X^*$ to $Y$. Finally, the transformation from source point cloud to target point cloud can be formulated as: $R = R_2 \cdot R_1$ and $t = R_2 \cdot t_1 + t_2$. We set top-k to 3 and 1 for training and testing.

### 3.3 DATA PRE-PROCESSING

The points of source and target point clouds are constant using the manner in RPMNet(Yew & Lee, 2020) for generating partial data. However, The number of points in the point cloud obtained by each lidar scan is not equal. To create a more realistic situation, a random-parameter is introduced to generate partial data in our paper. The random-parameter is denoted as $N_{keep} = random(w - \varphi, w + \varphi)$, where $w$ is the parameter for generating partial data in RPMNet and $\varphi$ is a random factor. However, there are some modules in which the number of points in source and target point clouds needs to remain the same, including our Transformer-based Overlapping-Score Prediction module. Methods of down-sampling may remove these overlapping points from the source or target point cloud. Pu-Net(Yu et al., 2018) proposes a method of up-sampling, which can transform the sparse point cloud into a dense point cloud. However, newly added points in Pu-Net will change the

Table 1: Results on ModelNet40, red indicates the best results and blue is the second-best among all methods.

| | Method | Error_R | Error_t | RMSE(R) | MAE(R) | RMSE(t) | MAE(t) |
|---|---|---|---|---|---|---|---|
| (a)Unseen Shapes | DCP(Wang & Solomon, 2019a) | 8.693 | 0.1174 | 6.501 | 4.547 | 0.0767 | 0.0584 |
| | RPMNet(Yew & Lee, 2020) | 1.175 | 0.0155 | 1.078 | 0.620 | 0.0174 | 0.0072 |
| | IDAM(Li et al., 2020) | 7.052 | 0.0864 | 7.469 | 4.937 | 0.0893 | 0.0615 |
| | RGM(Fu et al., 2021) | 3.147 | 0.0312 | 3.782 | 1.785 | 0.0386 | 0.0178 |
| | OMNet(Xu et al., 2021) | 1.118 | 0.0198 | 1.384 | 0.542 | 0.0226 | 0.0093 |
| | ROPNet(Zhu et al., 2021) | 0.852 | 0.0087 | 0.859 | 0.451 | 0.0085 | 0.0042 |
| | FINet(Xu et al., 2022) | 0.591 | 0.0110 | 1.267 | 0.269 | 0.0168 | 0.0048 |
| | RORNet(Wu et al., 2023c) | 2.035 | 0.0386 | 2.408 | 1.555 | 0.0425 | 0.0314 |
| | OGMM(Mei et al., 2023) | 1.209 | 0.0171 | 1.603 | 0.614 | 0.0265 | 0.0075 |
| | DIFT(Chen et al., 2023) | 0.472 | 0.0058 | 0.493 | 0.253 | 0.0072 | 0.0024 |
| | Ours | 0.392 | 0.0042 | 0.421 | 0.210 | 0.0053 | 0.0020 |
| (b)Unseen Categories | DCP(Wang & Solomon, 2019a) | 11.778 | 0.1402 | 8.556 | 6.134 | 0.0892 | 0.0701 |
| | RPMNet(Yew & Lee, 2020) | 1.333 | 0.0161 | 1.427 | 0.709 | 0.0161 | 0.0077 |
| | IDAM(Li et al., 2020) | 7.625 | 0.0881 | 8.152 | 5.193 | 0.0974 | 0.0680 |
| | RGM(Fu et al., 2021) | 4.291 | 0.0468 | 5.045 | 2.176 | 0.0503 | 0.0223 |
| | OMNet(Xu et al., 2021) | 3.206 | 0.0383 | 4.014 | 1.619 | 0.0406 | 0.0179 |
| | ROPNet(Zhu et al., 2021) | 1.042 | 0.0117 | 1.145 | 0.561 | 0.0145 | 0.0055 |
| | FINet(Xu et al., 2022) | 2.572 | 0.0311 | 3.918 | 1.286 | 0.0404 | 0.0142 |
| | RORNet(Wu et al., 2023c) | 2.742 | 0.0401 | 3.210 | 1.595 | 0.0452 | 0.0366 |
| | OGMM(Mei et al., 2023) | 1.915 | 0.0220 | 2.123 | 0.960 | 0.0308 | 0.0099 |
| | DIFT(Chen et al., 2023) | 0.643 | 0.0081 | 0.684 | 0.344 | 0.0121 | 0.0036 |
| | Ours | 0.489 | 0.0060 | 0.579 | 0.260 | 0.0088 | 0.0028 |
| (c)Gaussian Noise | DCP (Wang & Solomon, 2019a) | 12.341 | 0.1440 | 8.859 | 6.430 | 0.0912 | 0.0720 |
| | RPMNet(Yew & Lee, 2020) | 1.394 | 0.0175 | 1.336 | 0.741 | 0.0188 | 0.0083 |
| | IDAM(Li et al., 2020) | 8.786 | 0.1032 | 9.854 | 5.673 | 0.1154 | 0.0797 |
| | RGM(Fu et al., 2021) | 5.378 | 0.0578 | 5.763 | 2.810 | 0.0591 | 0.0280 |
| | OMNet(Xu et al., 2021) | 3.834 | 0.0476 | 4.356 | 1.924 | 0.0486 | 0.0223 |
| | ROPNet(Zhu et al., 2021) | 1.326 | 0.0145 | 1.328 | 0.709 | 0.0165 | 0.0070 |
| | FINet(Xu et al., 2022) | 2.984 | 0.0336 | 3.841 | 1.532 | 0.0379 | 0.0158 |
| | RORNet(Wu et al., 2023c) | 3.043 | 0.0588 | 3.585 | 1.604 | 0.0639 | 0.0424 |
| | OGMM(Mei et al., 2023) | 2.363 | 0.0279 | 2.172 | 1.220 | 0.0341 | 0.0125 |
| | DIFT(Chen et al., 2023) | 1.845 | 0.0198 | 2.135 | 1.076 | 0.0212 | 0.0095 |
| | Ours | 0.804 | 0.0091 | 1.023 | 0.434 | 0.0107 | 0.0044 |

original spatial distribution of the point cloud and cause correspondences that does not exist before. It is a principle in point cloud registration that newly added points cannot be used as corresponding points between the source and target point clouds. Therefore, a simple method is used in our paper:

- **Random Point**: The newly added points to source or target point cloud are selected from the same point cloud randomly. The new points have the same spatial coordinates as the original points in the point cloud, and will not change the spatial distribution of the point cloud, nor will they become corresponding points between source and target point clouds.

A certain amount of random points are added to the point cloud with fewer points. The random-parameter for generating partial data is used only in the test process.

## 3.4 Loss Functions

We adopt the loss which calculates the distance between the transformed point cloud and ground truth, and there are three in our model, including the initial transformation to ground truth, transformation to the newly added target point cloud and the final transformation to ground truth. For predicting overlapping scores, we adopt the cross-entropy loss function(Richard & Lippmann, 1991).

$$L_{\text{init}}^{\text{gt}} = \left\| X \cdot R_1^T - X \cdot R^{gt\,T} \right\|_1 + \left\| t_1 - t^{gt} \right\|_1, \quad L_{\text{init}}^{\text{new}} = \left\| X \cdot R_1^T - X \cdot R^{gt\,T} \right\|_1$$

$$L_{ol} = \frac{1}{2N} \sum_i \sum_j \left( S_{X^*}^{gt} \right)_{ij} \cdot \log \left( S_{X^*} \right)_{ij} + \frac{1}{2M} \sum_i \sum_j \left( S_Y^{gt} \right)_{ij} \cdot \log \left( S_Y \right)_{ij}, \tag{10}$$

$$L_{\text{finally}} = \left\| X \cdot R^T - X \cdot R^{gt\,T} \right\|_1 + \left\| t - t^{gt} \right\|_1,$$

where the $R^{gt}, t^{gt}$ denote the ground truth of rotation matrix and translation vector. $S_{X^*}^{gt}, S_Y^{gt}$ represent the ground truth of overlapping scores calculated as ROPNet. The finally loss can be calculated as:

$$L_{\text{total}} = \alpha L_{\text{init}}^{\text{gt}} + \beta L_{\text{init}}^{\text{new}} + \gamma L_{\text{finally}} + \delta L_{ol}, \tag{11}$$

where $\alpha, \beta, \gamma, \delta$ are set to 1 , 1, 1, 0.1 respectively.

## 4 EXPERIMENTS

In this section, we comprehensively evaluate our model on ModelNet40(Zhirong Wu et al., 2015), including the accuracy of registration and time consumption(Appendix A). We also prove that focusing on the rotation part is beneficial for point cloud registration.

### 4.1 DATASET

**ModelNet40**(Zhirong Wu et al., 2015) has been widely used for point cloud registration in recent years. With the in-depth study of semantic SLAM(Liao et al., 2022; Liu et al., 2022; Qian et al., 2022; Hu et al., 2022), the point cloud registration of semantic information is receiving more and more attention. There are 12311 CAD models in ModelNet40, one part for training, with 9843 models and the other is used for testing, with 2468 models. There are 40 categories in ModelNet40, including 8 symmetric categories, which are not suitable for registration. Therefore, these symmetric categories are removed in the test set as OMNet(Xu et al., 2021) and FINet(Xu et al., 2022). The partial point cloud is generated using the manner of RPMNet(Yew & Lee, 2020) and 30% of the points are removed. The random factor $\varphi$ is set to 0.05. We randomly generate three angles within $[0, 45]$ and translations within $[-0.5, 0.5]$ on the $x, y$ and $z$ axis respectively.

Table 2: Ablation Studies

|     | Initialization | TOSP | AFMR | Error_R | Error_t |
|-----|:---:|:---:|:---:|:---:|:---:|
| (1) | ✓ | ✓ | ✓ | 0.804 | 0.009 |
| (2) |   | ✓ | ✓ | 2.915 | 0.031 |
| (3) | ✓ |   | ✓ | 3.211 | 0.036 |

**Implementation Details.** We train for 1240 epochs using Adam(Kingma & Ba, 2017) with an initial learning rate of 0.0001. The learning rate is changed with the schedule of cosine annealing(Loshchilov & Hutter, 2017). A non-iterative manner is chosen in the training process, however, we iterate the attention-based mismatched-point removal module 4 times in testing process. To select representative points, $N_1$ is set to 448 and top-prob is set to 0.6 in the training process, and 0.4 for testing. In the Point-Feature Augmentation module, the number of neighbor points is set to 64.

### 4.2 EVALUATION ON MODELNET40

**Baseline Algorithms.** We compare our model to some state-of-the-art models: DCP, RPMNet, IDAM, RGM, OMNet, ROPNet, FINet, RORNet, OGMM, DIFT. The results of OMNet and FINet are chosen from FINet(Xu et al., 2022). We evaluate the registration in terms of the anisotropic rotation and translation errors: root mean squared error(RMSE), mean absolute error(MAE) and isotropic errors.

**Unseen shapes.** We first evaluate our model on the same categories, using 40 categories in the training set, and excluding 8 symmetric categories from the test set. We sample twice for the source and target point clouds. Table 1(a) shows the results where our model achieves state-of-the-art performance. Considering ICP and some of its variants cannot deal with the partially visible data, therefore, their results are not reported.

**Unseen categories.** We evaluate our model on different categories to test the generalization ability of our model. The first 20 categories are used for training, and the last 18 asymmetric categories are used for testing. Table 1(b) shows the results where all of these learning-based models achieve worse performance, because there are features these models have

Table 3: Results of rotation-preference

|     | $\alpha/\beta$ | Error_R | Error_t |
|-----|:---:|:---:|:---:|
| (a) | 1 | 1.05 | 0.0082 |
|     | 1.5 | 0.91 | 0.0073 |
| (b) | 1 | 2.69 | 0.0188 |
|     | 1.5 | 2.42 | 0.0176 |

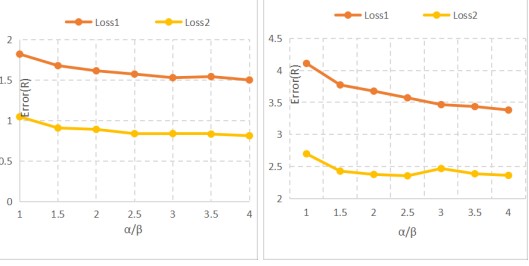

(a) Unseen shapes  (b) Unseen categories

Figure 3: Results of rotation-preference.

not seen before and this is exactly the downside of learning-based methods. However, our model still achieves the best performance among all methods, because our model combines global features using PointNet and local features using DGCNN in different modules. Our model not only learns the overall shape of the objects, but also learns the local geometric structure of the objects.

**Gaussian Noise.** We evaluate the robustness to noise of our model on the basis of the experiment Unseen categories. The noise is sampled from $N(0, 0.01^2)$ and clipped to $[-0.05, 0.05]$. As shown in Table 1(c), our model exhibits the best robustness, because of the choice of the weighted SVD algorithm based on overlapping scores. To evaluate the registration accuracy of our model at a large angle vividly, we set the three rotation angles to the maximum, and the registration results are shown in Fig. 4. More registration results are shown in Appendix B.

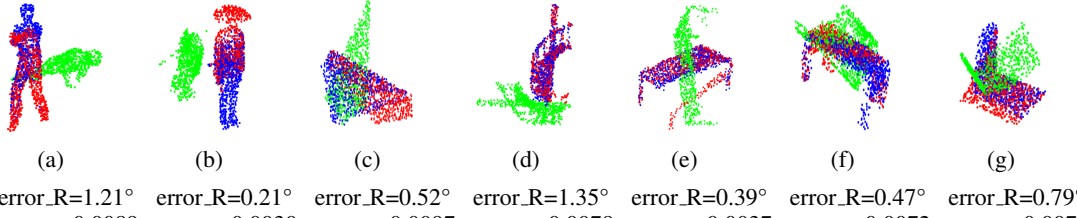

| (a) | (b) | (c) | (d) | (e) | (f) | (g) |
|---|---|---|---|---|---|---|
| error_R=1.21° | error_R=0.21° | error_R=0.52° | error_R=1.35° | error_R=0.39° | error_R=0.47° | error_R=0.79° |
| error_t=0.0089 | error_t=0.0030 | error_t=0.0097 | error_t=0.0078 | error_t=0.0037 | error_t=0.0072 | error_t=0.0079 |

Figure 4: Example results on ModelNet40. The green point cloud is the source point cloud, the red is the target point cloud, and the blue is the transformed source point cloud.

### 4.3 Ablation Studies

**Initialization.** We remove the initialization module to test the role of the initial transformation. As shown in Table 2, a good initial position is helpful for predicting overlapping points.

**Overlapping-Score.** In ablation studies, we evaluate the role of Transformer-based Overlapping-Score Prediction(TOSP) module. Comparing row 1 with 3 in Table 2, adding this module brings a great improvement to our model.

### 4.4 Rotation Preference

The aim of point cloud registration is to minimize the error of transformation from source point cloud to target point cloud. However, what determines the transformation accuracy is the error of the attitude between source and target point clouds. Furthermore, a smaller attitude error does not result in a larger translation error if models have been trained enough. We evaluate the effect of the rotation-preference model under two widely used loss functions:

$$
\begin{aligned}
Loss1 &= \alpha \cdot \left\| R^T \cdot R^{gt} - I \right\|^2 + \beta \cdot \left\| t - t^{gt} \right\|^2, \\
Loss2 &= \left\| X \cdot R^T - X \cdot R^{gt^T} \right\|_1 + \beta \cdot \left\| t - t^{gt} \right\|_1 + (\alpha - 1) \cdot \left\| X \cdot R^T - X \cdot R^{gt^T} \right\|_1.
\end{aligned}
\tag{12}
$$

**Implementation Details.** We use the model in PCRNet(Sarode et al., 2019), and train for 400 epochs using Adam with an initial learning rate of 0.0001. The learning rate is changed with the schedule of multi-step decay and the model is iterated 2 times. Given the simplicity of the model, we do not consider the partially visible data.

**Results.** As shown in Fig. 3, as the degree of rotation weight increases, the rotation error decreases continuously and only fluctuates in one case. A smaller rotation error does not result in a larger translation error, as shown in Table 3.

## 5 Conclusion

We design a dual-target model for partial-to-partial point cloud registration, which combines multiple features learned from PointNet, DGCNN, and attention module. By introducing a newly added target point cloud for registration, we verify that the rotation-preference loss function is beneficial for registration. We extensively use skip-connection and combine global features and local features, demonstrating that multi-level, multi-categories features are required in point cloud registration.

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

## A    APPENDIX

Considering that point cloud registration requires high real-time performance, so in this appendix, we evaluate the time consumption of our model. Besides, experiments with even smaller overlapping rates also show that our model only needs a few accurate overlapping points to achieve good results.

**Time Evaluation.** The task of point cloud registration has relatively high real-time requirements. There are many methods which run their models in a iterative manner. Although they achieve good performance, they also cause a lot of time consumption. Therefore, we also evaluate our model in a non-iterative manner. Table 4 shows that although the results running in non-iterative manner is worse than the iterative results, it is still better than other models. Besides, the non-iterative manner saves a lot of time.

Table 4: Unseen Shapes

|  | Method | Error_R | Error_t | time(ms) |
|---|---|---|---|---|
| Unseen Categories | ICP | 26.112 | 0.2001 | 26.7 |
|  | Ours | 0.489 | 0.0060 | 50.3 |
|  | Ours-v2 | 0.774 | 0.0098 | 30.3 |
| Unseen Shapes | ICP | 26.544 | 0.2045 | 21.4 |
|  | Ours | 0.392 | 0.0042 | 47.8 |
|  | Ours-v2 | 0.503 | 0.0055 | 28.2 |

We evaluate our model on GTX4090, ours-v2 denotes the results with non-iterative manner. We run ICP on cpu of Intel 12400. Considering that the operating environment is different, this is only for reference.

**Overlapping Rate.** The core idea of our model is to select some trustworthy overlapping points for accurate registration. Therefore, what determines the registration error is not the number of overlapping points, but whether the chosen overlapping points are trustworthy. Our model only needs a few overlapping points to achieve accurate registration. Experiments with even smaller overlapping rates are shown in Table 5. Partially visible data is generated using the manner of RPMNetYew & Lee (2020). And the $P_{keep}$ is the proportion of retained point clouds in RPMNet. $N_1$ is the number of overlapping points selected in the TOSP module. Although the error continues to increase as the point cloud retention ratio decreases, the results of our model are still acceptable.

Table 5: Method

|  | $N_1$ | Error_R | Error_t | RMSE(R) | MAE(R) | RMSE(t) | MAE(t) |
|---|---|---|---|---|---|---|---|
| $p_{keep} = 0.70$ | 448 | 0.804 | 0.0091 | 0.023 | 0.434 | 0.0107 | 0.0044 |
| $p_{keep} = 0.65$ | 416 | 0.938 | 0.0107 | 0.975 | 0.502 | 0.0120 | 0.0052 |
| $p_{keep} = 0.60$ | 354 | 1.402 | 0.0152 | 1.866 | 0.756 | 0.0164 | 0.0073 |
| $p_{keep} = 0.55$ | 352 | 2.802 | 0.0223 | 2.778 | 1.275 | 0.0222 | 0.0108 |
| $p_{keep} = 0.50$ | 320 | 3.485 | 0.0373 | 5.958 | 1.893 | 0.0469 | 0.0176 |

# B  APPENDIX

More results on ModelNet40 of Unseen Categories, The rotation angles of the x,y, and z axes are set to a maximum of 45 degrees.

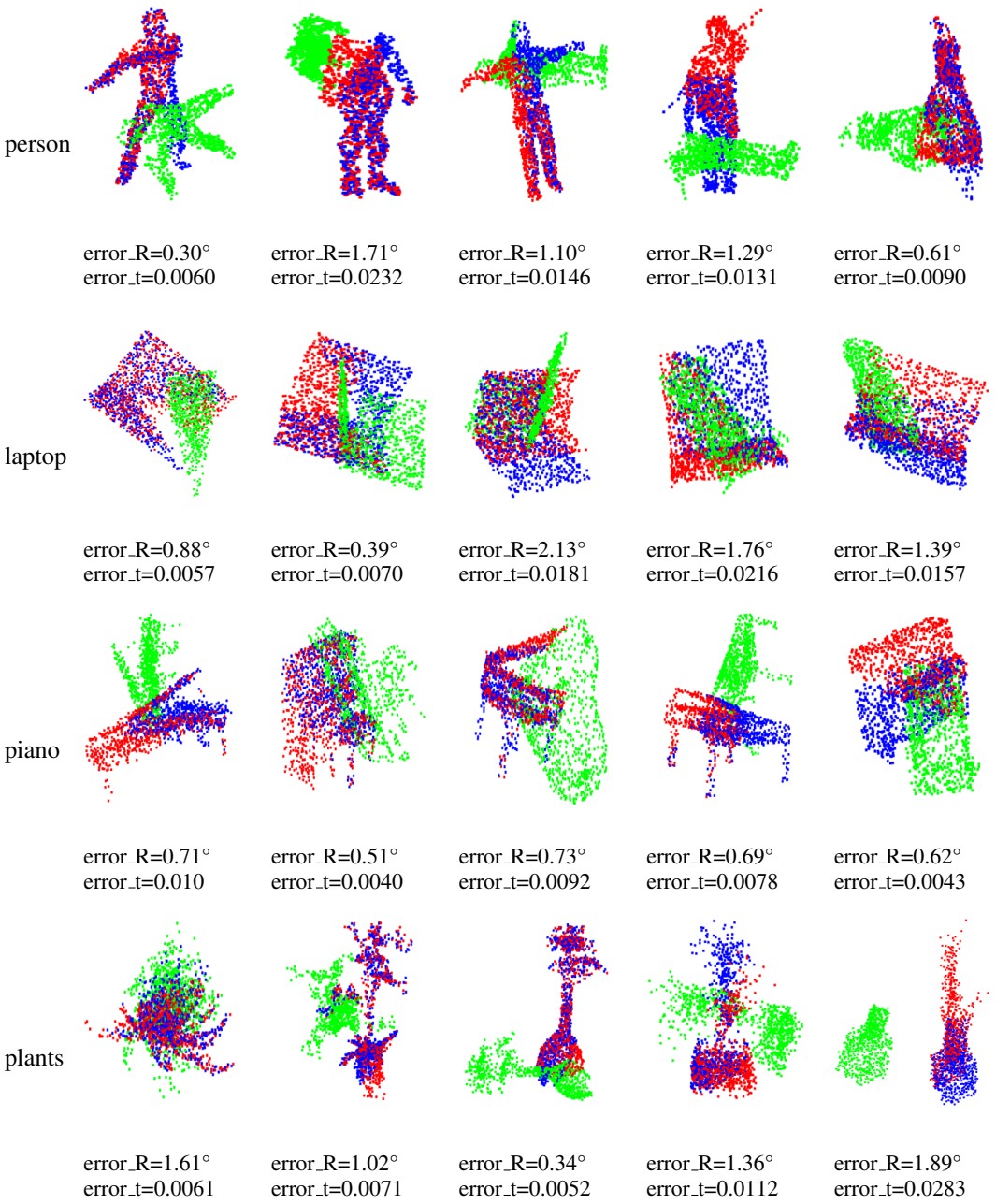

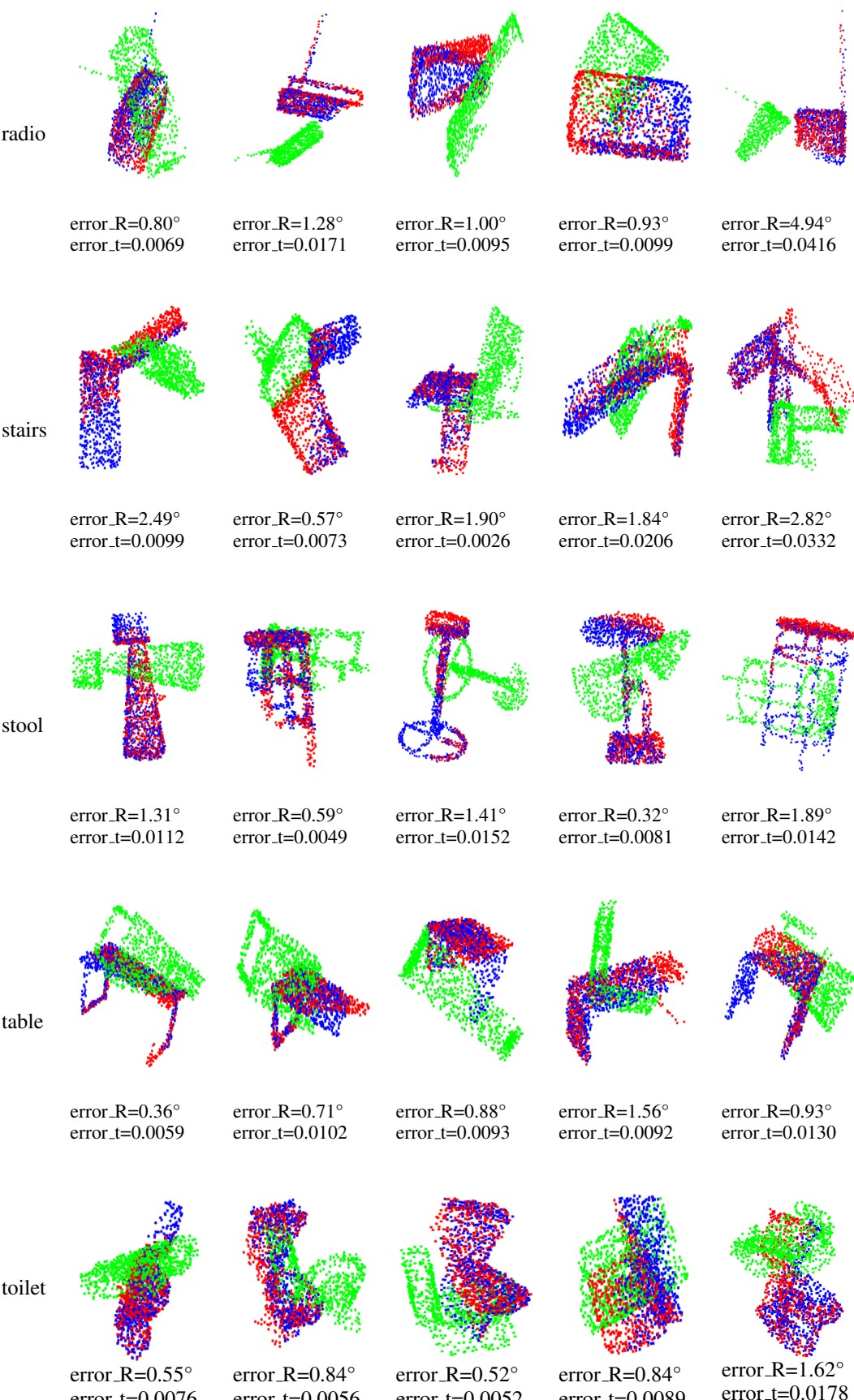

