# OpenReview forum: "Dual-target Point Cloud Registration Using Representative Overlapping Points"
_ICLR.cc/2024/Conference — ICLR 2024 Conference Withdrawn Submission_

### Official Review · Reviewer_hH1j · 2023-10-22

**Soundness:** 3 good
**Presentation:** 2 fair
**Contribution:** 2 fair
**Rating:** 3
**Confidence:** 5

**Summary:**

The paper proposes a dual-target point cloud registration model that combines multiple features learned from PointNet, DGCNN, and an attention module. The model aims to address the challenge of aligning partially visible point cloud data by introducing a new target point cloud obtained through rotation. The authors claim that their method achieves state-of-the-art performance compared to other learning-based methods on the ModelNet40 dataset.

**Strengths:**

1. This paper proposes a new perspective on point cloud registration by incorporating a dual-target model and emphasizing the alignment of attitude.
2. This paper combines multiple features learned from PointNet, DGCNN, and an attention module, allowing it to capture both global shape information and local geometric structures.

**Weaknesses:**

1. Lack of comparison with traditional methods.
This paper focuses on comparing the proposed method with other learning-based methods but does not provide a comparison with traditional point cloud registration algorithms [cite1-2].

[cite1] Zhou Q Y, Park J, Koltun V. Fast global registration[C]//Computer Vision–ECCV 2016: 14th European Conference, Amsterdam, The Netherlands, October 11-14, 2016, Proceedings, Part II 14. Springer International Publishing, 2016: 766-782.
[cite2] Yang H, Shi J, Carlone L. Teaser: Fast and certifiable point cloud registration[J]. IEEE Transactions on Robotics, 2020, 37(2): 314-333.


2. Experimental results are not convincing. This paper only conducts experiments on the small-scale and synthetic datasets. However, point cloud registration methods focusing on large-scale and real-world datasets[cite3, cite4] are more meaningful.

[cite3] Zeng A, Song S, Nießner M, et al. 3dmatch: Learning local geometric descriptors from rgb-d reconstructions[C]//Proceedings of the IEEE conference on computer vision and pattern recognition. 2017: 1802-1811.
[cite4] Shengyu Huang and et al., “Predator: Registration of 3d point clouds with low overlap,” in CVPR, 2021, pp. 4267–4276.

3. Lack of ablation studies. This paper needs to do some ablation studies to validate the effectiveness of each component in the proposed method. For example, investigating the impact of different hyperparameters or variations of the model architecture.

**Questions:**

1. Please compare with the traditional point cloud registration method[cite1-2].

2. Please conduct experiments on large-scale real-world datasets[cite3-4].

3. Please conduct ablation studies to validate the effectiveness of each component.

---

### Official Review · Reviewer_5tja · 2023-10-30

**Soundness:** 1 poor
**Presentation:** 2 fair
**Contribution:** 1 poor
**Rating:** 3
**Confidence:** 5

**Summary:**

This paper introduces a point cloud registration framework, in which a series of techniques are integrated. A coarse registration between the two point clouds is first performed, in which a second second target point cloud is generated to assist the initial registration. Then, in the fine registration step, overlap-estimation and mismatch filtering modules are designed to improve correspondences. Experiments are conducted on ModelNet 40 dataset.

**Strengths:**

The idea of generate a second target point cloud to assist registration is novel. The study of the relation between rotation and translation error is valuable.

**Weaknesses:**

1. The technical contribution is limited. The idea of generating a second target point cloud to assist registration is interesting but it is quite simple. Other techniques, such as overlap-ratio prediction and correspondence filtering have been widely used in this field, and specific comparison to similar components of the same purpose is needed to demonstrate the superiority of the designed models.

2. Experiments are conducted on one synthetic dataset, ModelNet40, is far from being enough. The performance on real and challenging datasets are needed.

3. There are some typos in the manuscript.

**Questions:**

1.At least, experiments on 3DMatch and 3DLoMatch are needed to demonstrate the applicability of the proposed method on real data.

2.Is the generated target point cloud used in the fine registration after initialization?

3.Can other baseline methods be used after the initialization step for fine registration? What would the results be?

4.It’s not clear how N_keep is used and why the partial point clouds are more realistic situation by using  N_keep. It is suggested to use the same way of generating experimental data if possible for easy and direct comparison with existing research.

5.“alignment of attitude” should be further explained since for easier understanding for it is not widely used in point cloud registration.

---

### Official Review · Reviewer_XE5g · 2023-10-31

**Soundness:** 1 poor
**Presentation:** 1 poor
**Contribution:** 1 poor
**Rating:** 1
**Confidence:** 5

**Summary:**

This paper proposes a dual-target model for point cloud registration, which combines features of PointNet and DGCNN. It first utilizes an initialization module for coarse registration, then uses a two-step attention-based representative overlapping-point selection module to determine the overlapping points. The experimental results demonstrate that the proposed method could perform well on ModelNet40.

**Strengths:**

1. High performance is achieved on ModelNet40 dataset.

**Weaknesses:**

1. This paper lacks novelty. For example, the initialization module is very similar to ROPNet, combining high-level and low-level features is not new. For the two-step attention-based representative overlapping-point selection module, the inter-transformer module is the same as the attention module in DCP, the Point-Feature Augmentation module is used following ROPNet, and the two-step process for selecting the representative overlapping points is still followed ROPNet. The overlap prediction is not new for point cloud registration task, many works have been done on it, such as Predator[1], Cofinet[2], Geotransformer[3], PEAL[4] and so on. I do not identify any significant differences between this paper and these approaches. Therefore, I think that this paper is not sufficiently innovative enough to be accepted by ICLR.
2. The paper is overall poorly written and it is a terrible reading experience. See Questions for specific issues. I think the writing of this paper needs a major improvement.
3. The authors utilize a Res-PointNet module to concatenate the output of each convolution block in the MLPs, why the outputs of fourth convolution block are not used? Is this a decision based on empirical evidence? If yes, I think giving some experimental verification is necessary.
4. The experimental results of this paper are only tested on ModelNet40. However, the widely adopted 3DMatch, 3DLoMatch and KITTI benchmarks should also be considered. I suggest that the authors should conduct more comprehensive and extensive experiments. So far, the performance of the proposed method is not convincing enough.
5. The ablation studies about the AFMR module is not given. And what is AFMR module? Is it the Attention-based Mismatched-point Removal?
[1] Huang S, Gojcic Z, Usvyatsov M, et al. Predator: Registration of 3d point clouds with low overlap[C]//Proceedings of the IEEE/CVF Conference on computer vision and pattern recognition. 2021: 4267-4276.
[2] Yu H, Li F, Saleh M, et al. Cofinet: Reliable coarse-to-fine correspondences for robust pointcloud registration[J]. Advances in Neural Information Processing Systems, 2021, 34: 23872-23884.
[3] Qin Z, Yu H, Wang C, et al. Geometric transformer for fast and robust point cloud registration[C]//Proceedings of the IEEE/CVF conference on computer vision and pattern recognition. 2022: 11143-11152.
[4] Yu J, Ren L, Zhang Y, et al. PEAL: Prior-Embedded Explicit Attention Learning for Low-Overlap Point Cloud Registration[C]//Proceedings of the IEEE/CVF Conference on Computer Vision and Pattern Recognition. 2023: 17702-17711.

**Questions:**

1. Confusing use of mathematical symbol. For example, the font formatting of cat(·) needs to be standardized. What is $F_{X^{∗}}$ and $F_{X^{∗}}^{g}$? What does the augmented feature $F_{m_{i}}^{L}$ means? And what is the $m_{i}$?
2. The authors state that “the transformation to the newly added target point cloud is formulated as $X^{'} = X · R_{1}$”. What is the $X^{'}$?
3. “Through this module, non-overlapping points will be removed and a new transformed source point cloud $X_{ro1}^{*}$ is obtained.” How to remove the non-overlapping points? And how to obtain the new transformed source point cloud?
4. What are the input and output of DGCNN?

**Details Of Ethics Concerns:**

None.